# Uncovering the Differences in Flavour Volatiles from Hybrid and Conventional Foxtail Millet Varieties Based on Gas Chromatography–Ion Migration Spectrometry and Chemometrics

**DOI:** 10.3390/plants14050708

**Published:** 2025-02-26

**Authors:** Zhongxiao Yue, Ruidong Zhang, Naihong Feng, Xiangyang Yuan

**Affiliations:** 1College of Agronomy, Shanxi Agricultural University, Taigu, Jinzhong 030801, China; 534265957@sxau.edu.cn; 2Institute of Industrial Crops, Shanxi Agricultural University, Taiyuan 030031, China; zhangruidong@sxau.edu.cn (R.Z.); fnaihong@126.com (N.F.)

**Keywords:** foxtail millet, volatile flavour components, gas chromatography–ion migration spectrometry, similarity analysis

## Abstract

The flavour of foxtail millet (*Setaria italica* (L.) P. Beauv.) is an important indicator for evaluating the quality of the millet. The volatile components in steamed millet porridge samples were analysed using electronic nose (E-Nose) and gas chromatography–ion mobility spectrometry (GC-IMS) techniques, and characteristic volatile fingerprints were constructed to clarify the differences in the main flavour substances in different foxtail millet varieties (two hybrids and two conventional foxtail millets). After sensory evaluation by judges, Jingu 21 (JG) scored significantly higher than the other varieties, and the others were, in order, Jinmiao K1 (JM), Changzagu 466 (CZ) and Zhangzagu 3 (ZZ). E-Nose analysis showed differences in sulphides and terpenoids, nitrogen oxides, organosulphides and aromatic compounds in different varieties of millet porridge. A total of 59 volatile components were determined by GC-IMS in the four varieties of millet porridge, including 23 aldehydes, 17 alcohols, 9 ketones, 4 esters, 2 acids, 3 furans and 1 pyrazine. Comparative analyses of the volatile components in JG, JM, ZZ and CZ revealed that the contents of octanal, nonanal and 3-methyl-2-butenal were higher in JG; the contents of trans-2-butenal, 2-methyl-1-propanol, trans-2-heptenal and trans-2-pentenal were higher in JM; and the contents of 2-octanone, hexanol, 1-octen-3-ol, 2-pentanone and butyraldehyde were higher in ZZ. The contents of 2-butanol, propionic acid and acetic acid were higher in CZ. A prediction model with good stability was established by orthogonal partial least squares discriminant analysis (OPLS-DA), and 25 potential characteristic markers (VIP > 1) were screened out from 59 volatile organic compounds (VOCs). These volatile components can be used to distinguish the different varieties of millet porridge samples. Moreover, we found conventional foxtail millet contained more aldehydes than the hybridised foxtail millet; especially decanal, 1-nonanal-D, heptanal-D, 1-octanal-M, 1-octanal-D and 1-nonanal-M were significantly higher in JG than in the other varieties. These results indicate that the E-Nose combined with GC-IMS can be used to characterise the flavour volatiles of different foxtail millet, and the results of this study may provide some information for future understanding of the aroma characteristics of foxtail millet and the genetic improvement of hybrid grains.

## 1. Introduction

Foxtail millet (*Setaria italica*) is the world’s sixth-largest cereal crop [1], highly resilient [2,3] and widely adapted [4] and widely grown in arid as well as semi-arid regions. The foxtail millet, after processing and shucking, is used to make porridge and soup, which are rich in proteins, fats, carbohydrates and trace elements [5] that are beneficial to the human body [6]. Foxtail millet porridge has therapeutic effects, such as stomach and digestion, anti-inflammatory and other pro-health effects. It has a long history in the medicine and food of Chinese food culture [7], and is loved by consumers. The advantages of hybrids have effectively increased the yield of millet, making it a high-yielding crop [8]. Hybrid millet showed significant hybrid dominance in terms of yield, drought tolerance and shelf life. The successful selection and breeding of hybrid foxtail millet is conducive to food security and the enrichment of highly processed products [9]. Foxtail millet is mostly consumed as a raw material for processed food. Eating quality is an important indicator for evaluating a cereal variety; poor eating quality is an important factor limiting the cultivation of hybrid foxtail millet on a large scale. Therefore, it is important to study and improve the eating quality of hybrid foxtail millet.

Aroma is the main indicator for evaluating the quality of millet, and millet with a strong aroma is more popular with consumers. Currently, the detection of volatile compounds in food has attracted considerable attention in the food industry. Gas chromatography–mass spectrometry (GC-MS), gas chromatography–olfactometry–mass spectrometry (GC-O-MS), electronic nose (E-Nose) and gas chromatography–ion mobility spectrometry (GC-IMS) have been widely used to analyse the flavour components in various types of foods. Liu et al. [10] investigated and compared the odour components of foxtail millet from a spring sowing area and a summer sowing area using gas chromatography–mass spectrometry (GC–MS) and simultaneous distillation extraction (SDE) and suggested that spring-sown millet showed higher diversity and contents of odour components compared to summer-sown millet. Zhang et al. [11] found that the cooking time and pH had an effect on the flavour compounds of different varieties of foxtail millet, which were determined using GC-MS analysis.

GC-IMS technology showed great advantages, such as speed, high resolution and ease of visualisation through simple sample preparation compared to GC-MS [12]. Simultaneously, these techniques are combined with chemometric methods such as principal component analysis (PCA), partial least squares discriminant analysis (PLS-DA) and orthogonal partial least squares discriminant analysis (OPLS-DA), which help to screen for different volatile components in different samples [13]. In recent years, a number of studies have reported differences in the flavour of different varieties of foxtail millet. Jin et al. [14] reported volatile flavour components of several coloured foxtail millets via gas chromatography–ion migration spectrometry and chemometrics. Kang et al. [15] analysed the volatile characteristic fingerprints of 20 foxtail millet cultivars and the differences in their volatile compounds by electronic nose (E-Nose), headspace gas chromatography–ion mobility mass spectrometry (HS-GC-IMS) and headspace solid-phase microextraction–gas chromatography–mass spectrometry (HS-SPME/GC-MS) techniques. However, little research has been published on the identification, fingerprinting and differential analysis of the full range of volatile organic components in hybrid and conventional foxtail millet using GC-IMS.

The objective of this study was to identify and quantify volatile compounds in different Chinese varieties of foxtail millet (two hybrids and two cultivars), using GC-IMS and E-Nose techniques. This study comprehensively investigated the differences in flavour between hybrid foxtail millet and conventional foxtail millet and provides a theoretical basis for the quality improvement of foxtail millet.

## 2. Materials and Methods

### 2.1. Materials and Reagents

All four millet varieties (Jingu 21, Jinmiao K1, Zhangzagu 3, Changzagu 466) were grown at the Industrial Crops Research Institute of Shanxi Agricultural University (37°15′ N, 111°47′ E) under the same environmental and field management conditions, and harvested in September 2023.

Jingu 21 (JG) is a popular high-quality millet, which was selected and bred by the Industrial Crops Research Institute of Shanxi Agricultural University. The millet, with a golden colour, strong aroma and edible quality, belongs to the first-class standard of high-quality foxtail millet [16].

Jinmiao K1 (JM) is a new high-quality drought-resistant and herbicide-resistant (enfluorfen) cereal variety selected by the Chifeng Scientific Research Institute of Agriculture and Animal Husbandry. This foxtail millet was rated as a first-class quality cereal grain in the 13th National Quality Edible Corn Selection in 2019 [17].

Zhangzagu 3 (ZZ) is a hybrid foxtail millet selected and bred by Zhangjiakou Academy of Agricultural Sciences. This foxtail millet variety, with a high yield, water conservation benefits, barren soil and drought resistance, wide adaptability and many other advantages, is considered an ideal crop for China’s arid areas [9].

Changzagu 466 (CZ) is a high-quality, high-yielding, herbicide-resistant hybrid foxtail millet suitable for cultivation in the mid-to-late-maturing regions of northwestern China, with a bright yellow colour, which was rated as having Grade 2 quality at the 12th Quality Millet Selection Meeting in 2017 [18].

The following materials were used in this study: Ortho ketones, 2-butanone, 2-pentanone, 2-hexanone, 2-heptanone, 2-octanone and 2-nonanone (all analytically pure), Aladdin Company, Bay City, MI, USA; 99.999% nitrogen; 20 mL headspace vial (Shandong Haineng Scientific Instrument Co Ltd., Dezhou, China); DB-WAX capillary chromatography column (15 m × 0.53 mm, 1.0 μm) (Restek, Bellefonte, PA, USA).

### 2.2. Cooking Procedures and Sensory Evaluation

First, 100 g of millet grains were accurately weighed. Then, 2000 mL of water, which was 20 times the volume of the millet, was added. Next, the millet and water mixture was placed in an induction cooker, and the output power was set to 2100 watts. The mixture was boiled for 30 min. We recruited 50 individuals aged between 40 and 65. These people participated in a training on the sensory evaluation of millet congee. Based on a strict and standardized evaluation process, they conducted a comprehensive evaluation of the millet congee using multiple key indicators, such as palatability and odour perception. The evaluation adopted a 100-point scoring system with a full score of 100. During the scoring process, each evaluator adhered to an objective and fair attitude and carefully assigned corresponding scores to each indicator based on their own sensory experiences.

### 2.3. Electronic Nose Analysis

The analysis of the volatile compounds in millet was carried out using a HERACLES II E-Nose (Alpha MOS, Toulouse, France). Eight grams of millet flour was incubated at 40 °C for 30 min in a 20 mL cell sealed with a magnetic cap. Then, 5 mL of headspace was withdrawn with a syringe and injected into the gas chromatography instrument. The split flow rate at the column heads was 10 mL/min, and hydrogen was used as the carrier gas. The volatile compounds were then separated on MXT-5 and MXT-1701 columns (20 m × 0.18 mm × 0.4 µm, Restek, Co., Bellefonte, PA, USA). The temperature of the two flame ionisation detectors was set to 250 °C and held for 10 s, then ramped to 250 °C at 1.5 °C/s.

### 2.4. Headspace–Gas Chromatography–Ion Mobility Spectrometry (HS-GC-IMS) Analysis

Volatile compounds in different varieties of foxtail millet were determined by 7610 GC-MS (Thermo Fisher Scientific Inc., Waltham, MA, USA) using a DB-5ms capillary GC column (30 m × 0.25 mm × 0.25 µm). For this, 5 g of millet sample was accurately weighed and placed in a 20 mL headspace flask, 7.5 mL of distilled water was added and the sample was boiled at 100 °C for 30 min, incubated at 80 °C for 15 min and then injected into the sample, and three parallel sets were determined for each sample. The headspace injection conditions were as follows: the incubation temperature was 80 °C, the incubation time was 15 min, the injection volume was 500 µL, non-shunt injection was employed, the incubation speed was 500 rpm and the injection needle temperature was 85 °C. The sample was injected into the headspace with a volume of 500 µL. The GC conditions were as follows: the column temperature was 60 °C, the carrier gas was high-purity helium (purity ≥ 99.999%) and the programmed ramp-up was an initial flow rate of 2 mL/min, held for 2 min, linearly increased to 10 mL/min within 8 min, then linearly increased to 100 mL/min within 10 min and held for 10 min. The running time of the chromatographic run was 30 min; the inlet temperature was 80 °C. The IMS conditions were as follows: the ionisation source was a tritium source (3H), the length of the migration tube was 53 mm, the electric field strength was 500 V/cm, the temperature of the migration tube was 45 °C, the drift gas was high-purity nitrogen (purity ≥ 99.999%), the flow rate was 150 mL/min and the positive ion mode was used. N-alkanes C7–C30 (Sigma-Aldrich, St. Louis, MO, USA) were used as external references to calculate the retention index (RI) of volatile compounds. Volatile compounds were tentatively identified by comparing the mass spectra and RI of the volatiles in the NIST 11 library. The concentrations of volatile compounds were determined by the internal standard method.

The calibration curves of the retention time and retention index were established by detecting the mixed standards of six ketones, and the retention indices of the substances were subsequently calculated from the retention times of the targets, which were qualitatively analysed by using the built-in GC retention indices (NIST 2020) database of VOCal software and the IMS migration time database for searching and comparing. Reporter, Gallery Plot and Dynamic PCA plug-ins in VOCal data processing software were used to generate 3D spectra, 2D spectra, difference spectra and fingerprints of volatile components, respectively, for the comparison of volatile organic compounds among samples.

### 2.5. Statistical Analysis

The experimental data were statistically analysed using SPSS 16.0 (IBM, Armonk, NY, USA); radar plot analysis was performed using Origin Pro 8.0 software (Origin Lab Inc., Northampton, MA, USA); clustered heat map analysis was performed using GraphPad Prism software (v8.0 version). Statistical differences between the samples were determined by one-way analysis of variance (ANOVA) and Duncan’s Multiple Polar Difference tests, and the results were expressed as mean ± standard deviation; OPLS-DA and Projected Importance (VIP) were used to perform the multivariate statistical analysis by SIMCA 14.1. Principal component analysis (PCA) and heat mapping were performed using Metware Cloud, a free online platform for data analysis (https://cloud.metware.cn).

## 3. Results and Discussion

### 3.1. Sensory Evaluation and Electronic Nose Analysis of Flavours in Four Foxtail Millet Samples

The four varieties of foxtail millet (JG, JM, CZ, ZX) were cooked into porridge, evaluated and scored by 50 assessors of aroma, and the results are presented in Appendix A. JG had the highest score among the four varieties, followed by JM, CX and ZX, respectively. The response values of the E-Nose sensor to the volatile flavour components of different varieties of foxtail millet are shown in Figure 1A, with significant differences in sensors W1W, W5S and W2W, indicating that there are differences in sulphides and terpenes, nitrogen oxides, organosulphides and aromatic compounds in different varieties of millet porridge. The peak areas of volatile components were used for the principal component analysis of the corresponding data from the E-Nose (Figure 1B) to clarify the differences in volatile components among cereal varieties. The cumulative contribution of the PCA model was 93.2%, indicating that the developed model reflects the overall information of the samples better. The different samples in the figure are distributed in different regions of the coordinate system, separated significantly without overlap, indicating that there are significant differences in flavour in different varieties of millet porridge. The ZZ and CZ samples were placed on the right side of the x-axis of PC1 (ZZ and CZ were classified in the first and fourth quadrants, respectively), and the JG and JM samples were placed on the left side of the x-axis of PC1 (JG and JM were classified in the second and third quadrants and partially clustered on the x-axis), suggesting that hybrid and conventional millets were well differentiated from each other. These results indicate that there are significant differences in the volatile components of different millet varieties, with specificity for each variety.

### 3.2. Comparative GC-IMS Analysis of Volatile Components in Millet Samples

The volatile organic components of conventional and hybrid millets were examined using GC-IMS, and 3D topographical maps were created based on the signal strength of each compound (Figure 2A). The three axes represent migration time (x-axis), retention time (y-axis) and signal peak intensity (z-axis), each point is a volatile organic compound, and different colours represent signal strength [19].

For a visual comparison of the differences in flavour components of different varieties of foxtail millet, Figure 2B was downscaled using a two-dimensional topological map. In the two-dimensional spectrogram, the background of the whole graph is blue, the vertical coordinate represents the retention time (s) of the gas chromatogram, and the horizontal coordinate represents the relative migration time of the ions (normalized); the red vertical line at 1.0 of the horizontal coordinate is the RIP (Reaction Ion Peak, normalized); each point on both sides of the RIP represents a kind of volatile organic compound (VOC), and the colour of the peak represents the signal intensity of the VOC. Each point on both sides of the RIP represents a volatile organic compound, and the colour represents the intensity of the peak signal of the volatile organic compound, ranging from blue to red, with the darker colour indicating the greater intensity of the peak. In order to further visually compare the composition and differences in volatile components in conventional and hybrid millet, the spectrum of JG was selected as a reference, and the remaining samples’ spectra were sequentially deducted from the reference to obtain a difference comparison graph (Figure 2C). By comparing the spot intensities of volatile organic compounds of four different samples, the VOCs of conventional and hybrid grains showed significant differences in the GC-IMS characteristic spectra, and the aroma components could be well distinguished. The signal intensities of the VOC peaks of different varieties of millet showed an increasing, decreasing, disappearing or fluctuating state, indicating that the types of major aroma components were different among different varieties, and the relative contents showed a tendency of increasing and decreasing differences. The differences in the volatile aroma components of different millet porridges may be related to their nutritional and physicochemical components, fatty acids, polyphenol flavonoids and other content [15], which may be oxidised and cleaved during the heating process of high-temperature cooking porridges and, thus, cause the differences in the volatile components of different varieties.

### 3.3. Qualitative Analysis of Volatile Component Profiles

The volatile components in the millet were determined by the GC-IMS technique, and the characterisation of volatile odours was performed using NIST and IMS databases [20]. The volatile compounds were identified using Laboratory Analytical View (LAV) software and a GC-MS library search by comparing their retention indices (RIs) and drift times (DTs) with those of the standard n-ketones C4~C9 (Sinopharm Holding Chemical Reagent Beijing Co., Ltd., Beijing, China) in the IMS and in-built databases [21,22]. The volatile content is proportional to the peak intensity, and each point marked by a number represents a characterised volatile component; the qualitative spectrum is shown in Figure 2A. A total of 59 volatile components (monomers and dimers) were determined in the four samples, and the details of the volatile components are shown in Table 1, including 23 aldehydes, accounting for 38.98%, 17 alcohols, accounting for 28.81%, 9 ketones, accounting for 15.25%, 4 esters, accounting for 6.80%, 2 acids, accounting for 3.39%, 3 furans, accounting for 5.08% and 1 pyrazine, accounting for 1.69%. It can be seen that the volatile components in millet porridge are mainly aldehydes, alcohols, ketones and esters (Appendix A), which is consistent with the results of Liu’s [23] study.

### 3.4. Comparative Analysis of the Fingerprints of Volatile Components from Four Different Foxtail Millet Varieties

In order to more visually show the differences in the volatile organic compositions of millet porridge from conventional and hybrid grains, the volatiles were further compared in the millet after cooking, all the flavour matter spots in the GC-IMS profiles were measured, and the fingerprints were used to visualise and analyse all the volatiles from the four varieties (Figure 3). Each row of the graph represents all the signal peaks selected in one sample, and each column represents the signal peaks of the same VOC in different samples. As can be seen in Figure 3, the complete VOC information for each sample is significantly different due to the corresponding signal intensity. Several VOCs with high signal intensity were identified in all the samples, which were 2-acetone, ethanol, pentanol, 1-penten-3-ol, hexanal, pentanal and propanal, which are well-known aromatic compounds. Among them, the signal intensities of octanal (monomer and dimer), nonanal (monomer and dimer) and 3-methyl-2-butenal (monomer and dimer) in JG were significantly higher than those of the other samples, suggesting that the content of these substances was higher. Trans-2-butenal, 2-methyl-1-propanol, trans-2-heptenal and trans-2-pentenal are high in JM. Substances present at higher levels in ZZ than in the other three samples include 2-octanone, hexanol, 1-octen-3-ol, 2-pentanone, butyraldehyde, etc. The highest signal intensities were found for 2-butanol, propionic acid and acetic acid in the CZ. This is consistent with the results of the E-Nose analysis.

Aldehydes are mainly produced by the oxidative breakdown of lipids and are oxidation products of polyunsaturated fatty acids [24]. Aldehydes are the most abundant group of volatile components in grains and contribute the most to the overall volatile flavour of the grain [25,26]. Simple aldehydes have a pungent odour, and trace aldehydes can make the aroma more mellow; other aldehydes have a lower threshold of aromatic odour and volatility. The aroma of the grain cooking process has a significant impact on the overall aroma; generally, it has a grassy aroma, as well as floral, fresh, fruity, creamy, fatty, green grassy and other odours. The formation of these volatiles is the main contributor to the flavour of the aromatic grain [27]. A total of 23 aldehydes were detected in conventional and hybrid grains in this experiment, including benzaldehyde, nonanal, octanal, 3-methyl-2-butenal, trans-2-butenal, trans-2-heptenal and trans-2-pentenal. Among them, hexanal, pentanal and propanal were aromatic compounds common to both conventional and hybrid grains, while the signal intensity of aldehydes such as octanal (monomer and dimer), nonanal (monomer and dimer) and 3-methyl-2-butenal (monomer and dimer) was significantly higher in JG than in ZZ.

The odour of both octanal and nonanal is mainly due to the oxidative degradation of hydroperoxides of oleic acid [28]. Of these, nonanal has a strong, greasy odour, with grassy and sweet orange notes, while octanal has an odour that gives a floral contribution to the grain [29]. 3-Methyl-2-butenal, trans-2-heptenal and trans-2-pentenal belong to the group of 2-enals, and are typically a product of the Melad reaction or a thermal degradation product of sugars [30]. However, aldehydes increase in odour threshold with increasing c-chain length [31], with a decrease in citrus flavours and an increase in fruity and fatty flavours.

A total of 17 alcohols were detected, mainly monomers and some dimers and polymers, such as 1-octen-3-ol, 1-hexanol, 3-methyl-1-butanol, 2-methyl-1-propanol and so on. Alcohols are usually produced by the decomposition of secondary hydroperoxides of fatty acids, while alcohols are also precursors for the generation of aldehydes, ketones, esters, etc., under high-temperature conditions [32]. The saturated alcohols had a high threshold and did not contribute much to the aroma of the millet porridge; the unsaturated alcohols had a low threshold and might contribute to the aroma profile of millet porridge. The alcohols identified in the four samples were mainly hexanol and 1-octen-3-ol in ZZ and 2-butanol in CZ. Of these, hexanol has a light green leafy nose with hints of wine, fruit and fat [33]. 1-octen-3-ol is believed to be a hydroperoxide degradation product of linoleic acid [34], which, in high levels, gives millet porridge a mushroom aroma [35]. Orthohexanol has a light, greenish, tender, leafy aroma with slight winey, fruity and fatty notes.

Ketones are mainly produced from reactions such as the oxidation of alcohols and autoxidative decomposition of fatty acids, especially unsaturated fatty acids, or the degradation of amino acids, and they have a significant effect on the aroma characteristics of foods [36,37]. They generally have a fresh creamy flavour and fruit and vegetable aromas, and play a role in the formation of the aroma of millet porridge. A total of nine ketones were detected in this study, among which ZZ was relatively rich in ketones, including 2-octanone and 2-pentanone. Ketones have a lower aroma threshold and a higher ability to impart aroma to millet porridge. Previous studies have shown that butanone can confer chocolate and burnt flavours to millet porridge [38]; 2-heptanone is produced by the oxidation of linoleic acid and adds a fruity aroma to the samples [39].

The unique flavour of millet porridge is closely related to the type and content of esters, which are generated during yeast fermentation through esterification reactions between alcohols and fatty acids [40], or fatty acid and amino acid degradation products through esterification [41], and usually confer the substance with unique fruity and floral flavours [37]. Among them, ethyl acetate, which presents pineapple flavour, is an important aromatic compound and major fruit ester, which is formed by the esterification of fatty acids with acetic acid formed by the β-oxidation of ethanol [42]. Four esters were detected in this study, mainly ethyl-2-oxopropanoate, 2-acetic acid-1-methyl methyl ester, ethyl acetate, and ethyl caproate. Among them, ethyl acetate and ethyl-2-oxypropionate were abundant in JG, and these esters had the effect of enhancing the aroma, which combined with aldehyde volatile compounds to give JG a unique aroma.

Acids make up a small proportion of the sensory quality and usually have relatively high flavour thresholds [43,44], and, therefore, may have less influence on the odour of millet porridge. Although propionic and acetic acids were identified and abundantly concentrated in CZ, the aroma was not as ricey as that of conventional grains. Heterocyclic compounds are associated with baking and nutty flavours. Furans are derived from dehydrated carbohydrates (glucose pyrolysis) or the oxidation of fatty acids, or are produced by the Amadori rearrangement mechanism [45]. 2-pentylfuran has a typical nutty flavour at low concentrations, while at high concentrations, it imparts a soybean odour [34]; it has been identified as an important odorant in rice [46], and has the potential to have an effect on the aroma profile of millet porridge.

The composition of volatile compounds in different varieties of millet porridge was relatively similar, but the contents varied significantly. Aldehydes, alcohols, ketones, esters, enol terpenoids, furans and pyrazines contributed to the formation of the aroma to varying degrees, and differences in the content and type of VOCs led to differences in the aroma characteristics of the millet, and the millet porridge VOCs were usually a synergistic overall presentation of a variety of organic compounds.

### 3.5. Resemblance Analysis of Volatile Compounds of Different Foxtail Millet Varieties Through PCA

PCA has been used to analyse and compare the complex variables and categorize distinctness in odour compounds in different coloured wheat grains [47] and different coloured foxtail millet varieties [14]. Currently, PCA visually discriminates differences in the distribution of volatile compounds between conventional and hybrid grains. Millet samples with similar volatile compound profiles were clustered, whereas millet samples with different volatile components were separated on the score plot. As shown in Figure 4, the four foxtail millet samples were clustered into four different groups based on PC1 and PC2 scores in the dimensionality plot, with PC1 contributing 60.62%, PC2 contributing 22.72%, and the cumulative PC1 and PC2 contributing 83.34%, which can well represent the differential characteristics of the original variables. We can clearly see that the four samples of JG, JM, ZZ and CZ can be clearly distinguished from each other, in which ZZ is the farthest away from the other three samples, and JG, JM and CZ are relatively close to each other, indicating that there is a big difference in flavour between ZZ and the other three varieties; meanwhile, JM and JG can be clearly distinguished from ZZ and CZ, which indicates that the analysis results can better discriminate and distinguish between the conventional grains and the hybrids. This result is consistent with the results of fingerprinting. Fan et al.’s study also showed the identification of flavour volatiles of different colours of sorghum based on GC-IMS and PCA [22].

### 3.6. Orthogonal Partial Least Squares Discriminant Analysis with Cross-Validation

PLS-DA is a discriminant multivariate analysis method based on a model of the relationship between target components and sample categories, where the goodness-of-fit parameter (R^2^X) and the model explanatory power (R^2^Y) denote the explanatory variability of the constructed model for the X and Y matrices, respectively. The predictive power (Q^2^) denotes the predictive power of the model; a value closer to 1 implies that the model has a better generalised explanatory power, and the difference between the two should not be large [48]. OPLS-DA was used to classify the VOCs in different varieties of millet porridge. The results are shown in Figure 5. The model has values of R^2^X = 0.986, R^2^Y = 0.999 and Q^2^ = 0.0.998; this shows that the model is stable and predictive, and it can describe most of the data. The samples of different varieties of millet porridges are well differentiated in the scatter plot of the OPLS-DA scores, the classification effect is consistent with the PCA and the OPLS-DA model can further exclude irrelevant differences and achieve better separation of VOCs in samples between different groups. To avoid overfitting, the reliability of the OPLS-DA model was statistically validated using the replacement fit test. As shown in Figure 5B, after 200 cross-modelling validations, R^2^ intersects the vertical axis (0, 0.231) and Q^2^ intersects the vertical axis (0, −0.100), the values R^2^ and Q^2^ generated by the random arrangement on the left side are lower than those of the original data on the right side, and the regression line of Q^2^ intersects with the horizontal coordinates with the negative intercept, which indicates that there is no overfitting phenomenon in this OPLS-DA model, i.e., the constructed OPLS- DA model is stable and reliable, and can be used to discriminate and distinguish the volatile aroma components of different varieties of millet porridge.

### 3.7. Screening for Differential VOCs in Samples of Different Varieties of Millet Porridge

After visualising the 59 VOCs in the different varieties of millet porridge samples by GC-IMS fingerprinting, the contribution of each variable to the classification was quantified on the basis of variable importance in projection (VIP) based on the construction of a reliable OPLS-DA model to further identify the parameters that differentiate between conventional and hybridised grain samples [49]. The results are shown in Figure 6A; VOCs with VIP values greater than 1 are usually screened as differential flavour components between different samples of millet porridge. In this study, a total of 25 VOCs with VIP values greater than 1 were found in different varieties of millet porridge samples, including Ethanol-M, 1-nonanal-D, 1-Octen-3-ol-D, n-Pentanal, (E)-2-Pentenal-M, 1-Octen-3-ol-M, Decanal, Ethanol-D, 1-nonanal-M, (2.6)-dimethylpyrazine, (E)-2-Pentenal-D, (E)-2-hexen-1-al-M, 2-Butanone, 1-octanal-D, 2-methyl-1-propyl acetate, Heptaldehyde-D, (E.E)-2.4-heptadienal, 1-Pentanol-D, 1-hexanal, (E)-2-Heptenal-M, (Z)-furan linalool oxide-D, 3-Methyl butanal, 1-Penten-3-ol, 1-Pentanol-M and 1-octanal-M. Ethanol-M had the highest VIP value (1.2175), followed by 1-nonanal-D and 1-Octen-3-ol-D, with VIP values of 1.21 and 1.17, respectively. In this study, Ethanol-M was significantly higher in hybrid foxtail millet (ZZ and CZ) than in conventional foxtail millet (JZ and JM); however, the saturated alcohol threshold was high and did not contribute much to the flavour of the millet porridge. These substances were identified as key aroma compounds capable of explaining the aroma profile of the four varieties of millet porridge. In addition, PCA and heat map collections were conducted using these identifying indicator chemicals (Figure 6B,C). PCA allowed the isolation of most of the characters of foxtail millet with a reasonable total variance of 86.47% (67.48% and 18.99% for the first two components, respectively) (Figure 6B). The results of the clustered heat map showed that the 25 sieved odour chemicals in the four foxtail millet species better classified the differences between the cooked foxtail millet samples (Figure 6C). Moreover, aldehydes were significantly higher in JG than in the other varieties, especially decanal, 1-nonanal-D, heptanal-D, 1-octanal-M, 1-octanal-D and 1-nonanal-M. Wang et al. [50] found that the differentially labelled volatile components screened using OPLS-DA in combination with VIP > 1 could distinguish well between two kinds of honey. Jin et al. [14] also observed that OPLS-DA screening of differentially labelled odour components coupled with VIP values above 1.0 could classify four coloured foxtail millet species. Therefore, it is feasible to screen for differentially volatile flavour compounds from four foxtail millet species based on the OPLS-DA model and VIP values.

## 4. Conclusions

After cooking four foxtail millet varieties (two hybrids and two conventional foxtail millets), 59 odour chemicals were measured by the gas chromatography–mass spectrometry (GC-IMS) method. Aldehydes, alcohols and ketones were the main chemical types. The overall data and similarity analysis performed by GC-IMS enabled a good classification of the changes in volatile flavour compounds in the four foxtail millet varieties after cooking. A reliable predictive model was developed by OPLS-DA, and 25 odour markers (VIP > 1) were selected to characterise the four foxtail millet species after cooking. A clustered heat map revealed that conventional foxtail millet contained more aldehydes than the hybrid foxtail millet. After sensory evaluation by the jury, JG scored the highest, while the other varieties were JM, CZ and ZZ, in that order. The E-Nose analysis showed that there were differences in sulphides and terpenoids, nitrogen oxides, organic sulphides and aromatic compounds in different varieties of millet porridge. These results could be useful for future insights into flavour volatiles in foxtail millet and for the genetic improvement of hybrid foxtail millet.

## Figures and Tables

**Figure 1 plants-14-00708-f001:**
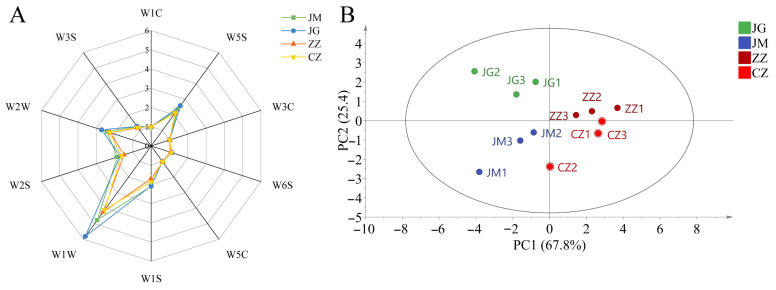
E-Nose sensor response radar plot (**A**) and principal component score plot (**B**) of sample foxtail millets. Notes: W1C responds to aromatic compounds, W5S to nitrogen oxides, W3C to amines and aromatic compounds, W6S to hydrides, W5C to alkanes and aromatic compounds, W1S to methane, W1W to sulphides and terpenes, W2S to alcohols and certain aromatic compounds, W2W to organic sulphides and aromatic compounds and W3S to alkanes.

**Figure 2 plants-14-00708-f002:**
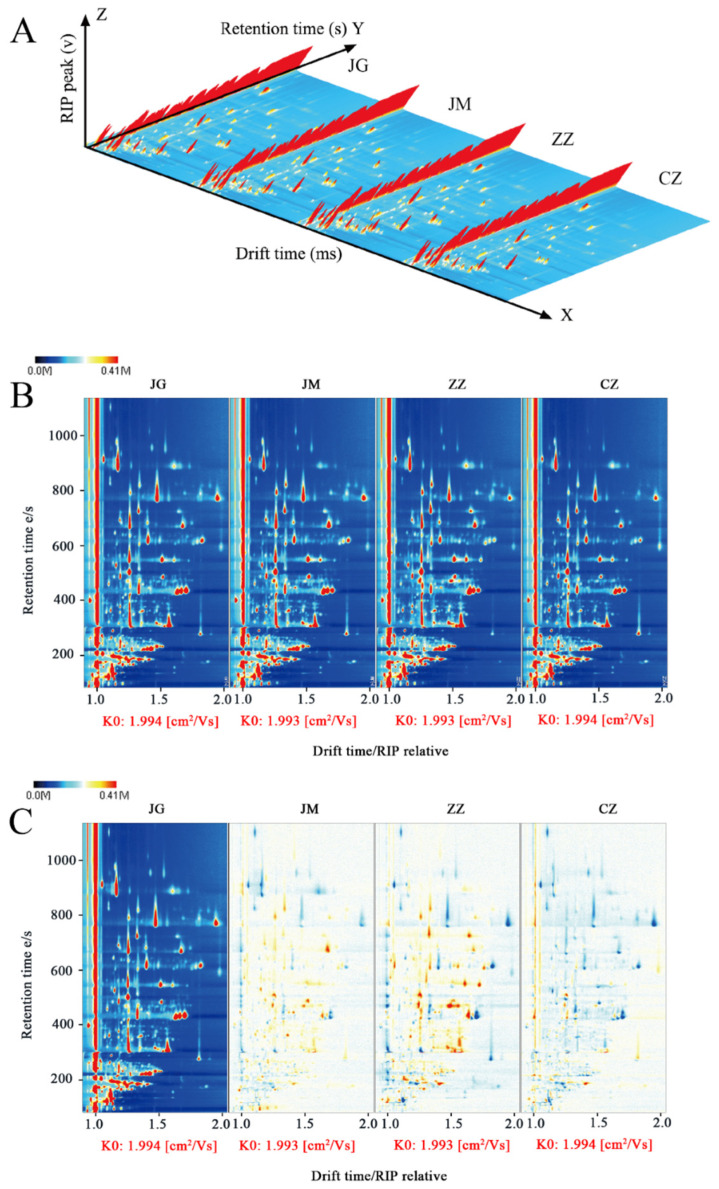
Three-dimensional topographic (**A**), two-dimensional spectra of volatile components (**B**) and contrast maps (**C**) of GC-IMS in different varieties of foxtail millet (JG, JM, ZZ and CZ). The red colour indicates the signal intensity, and each spot denotes one specific volatile component.

**Figure 3 plants-14-00708-f003:**
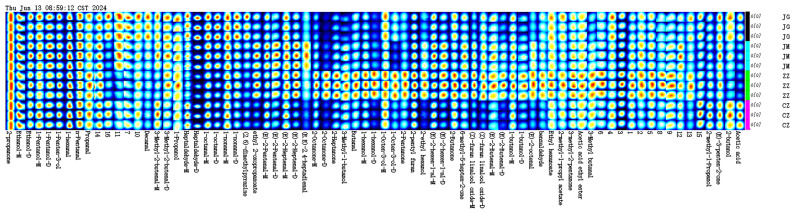
Fingerprints of flavour-imparting organic chemicals in four foxtail millet varieties.

**Figure 4 plants-14-00708-f004:**
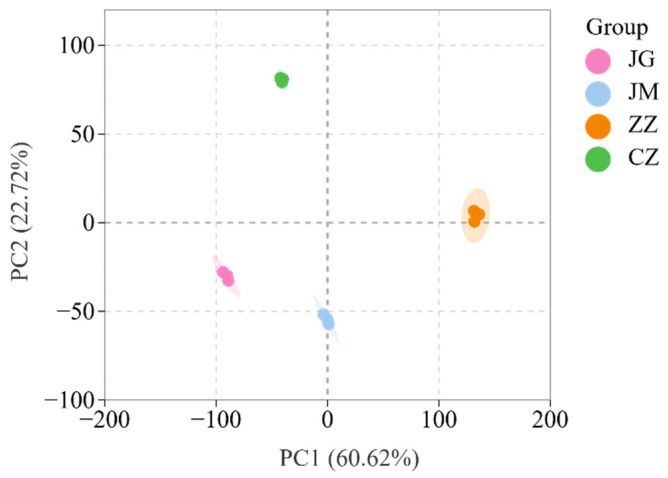
Comparison of volatile components in different foxtail millet varieties by PCA score plot.

**Figure 5 plants-14-00708-f005:**
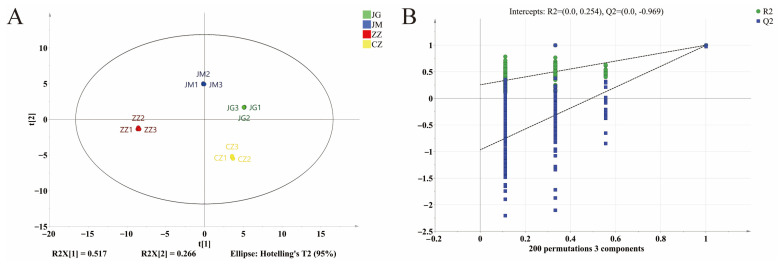
OPLS-DA scatter plot (**A**) and trans-verifications by a rearrangement trial (**B**) of odour profiles in four foxtail millet varieties.

**Figure 6 plants-14-00708-f006:**
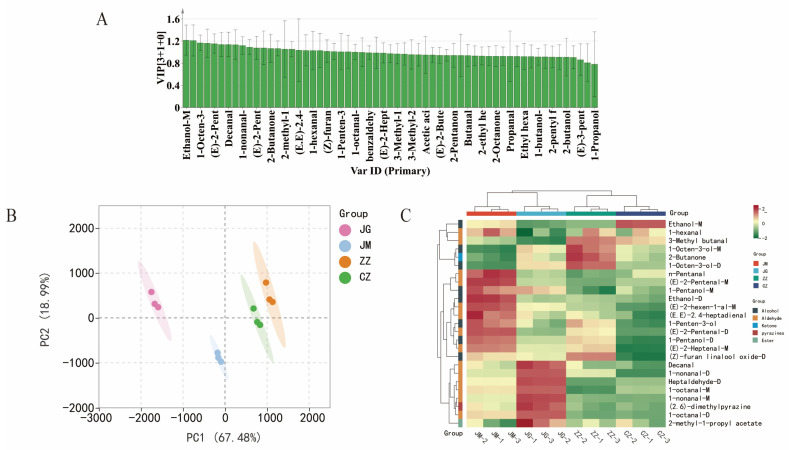
Sieving of differential volatile compounds in four foxtail millet varieties after cooking. (**A**) VIP distribution, (**B**) PCA scatter map, (**C**) clustering heat map.

**Table 1 plants-14-00708-t001:** Volatile organic compounds (VOCs) recognized from different foxtail millet varieties.

NO.	Chemical	Retention Index	Retention Time/s	Drift Time/ms	Relative Amount/%			
					JG	JM	ZZ	CZ
1	benzaldehyde	1504	976.529	1.15543	0.50 ± 0.02 b	0.49 ± 0.02 b	0.60 ± 0.03 a	0.42 ± 0.02 c
2	Acetic acid	1476.1	915.682	1.05313	1.81 ± 0.01 b	1.18 ± 0.03 c	0.82 ± 0.00 d	2.29 ± 0.02 a
3	1-Octen-3-ol-D	1464	890.354	1.16293	0.74 ± 0.00 b	0.35 ± 0.01 d	0.88 ± 0.02 a	0.57 ± 0.01 c
4	1-Octen-3-ol-M	1464	890.354	1.60547	4.33 ± 0.07 a	3.51 ± 0.01 b	4.30 ± 0.11 a	4.38 ± 0.05 a
5	(E)-2-octenal	1429.9	822.736	1.33719	0.33 ± 0.01 c	0.51 ± 0.01 b	0.59 ± 0.02 a	0.35 ± 0.00 c
6	1-nonanal-D	1400.8	769.199	1.47517	3.04 ± 0.03 a	2.25 ± 0.02 b	2.13 ± 0.01 c	1.89 ± 0.02 d
7	1-nonanal-M	1401.1	769.844	1.94523	4.86 ± 0.04 a	4.24 ± 0.05 c	3.80 ± 0.02 d	4.43 ± 0.04 b
8	1-hexanol-D	1375.4	725.311	1.32949	0.26 ± 0.00 c	0.37 ± 0.01 b	0.79 ± 0.01 a	0.27 ± 0.01 c
9	1-hexanol-M	1375.4	725.311	1.64222	1.36 ± 0.01 c	1.49 ± 0.01 b	2.20 ± 0.01 a	1.49 ± 0.01 b
10	6-methyl-5-hepten-2-one	1356.9	694.954	1.17902	0.53 ± 0.02 b	0.63 ± 0.02 a	0.61 ± 0.03 a	0.51 ± 0.01 b
11	(E)-2-Heptenal-D	1341.6	670.82	1.25746	2.06 ± 0.02 c	3.07 ± 0.03 a	2.86 ± 0.02 b	1.44 ± 0.01 d
12	(E)-2-Heptenal-M	1341.8	671.11	1.67101	3.32 ± 0.03 b	3.54 ± 0.02 a	3.15 ± 0.05 c	3.04 ± 0.04 d
13	(2.6)-dimethylpyrazine	1328.1	650.162	1.14275	0.26 ± 0.01 a	0.18 ± 0.01 b	0.13 ± 0.01 d	0.15 ± 0.01 c
14	1-octanal-D	1308.2	620.88	1.40205	1.81 ± 0.02 a	1.46 ± 0.01 b	1.03 ± 0.03 d	1.21 ± 0.02 c
15	1-octanal-M	1308.2	620.88	1.82503	3.16 ± 0.03 a	2.72 ± 0.03 b	2.08 ± 0.03 c	2.71 ± 0.03 b
16	1-Pentanol-D	1268.1	550.058	1.25548	2.21 ± 0.02 b	2.44 ± 0.01 a	2.13 ± 0.01 c	2.07 ± 0.01 d
17	1-Pentanol-M	1267.3	548.61	1.51143	3.45 ± 0.01 b	3.40 ± 0.02 c	2.88 ± 0.03 d	3.49 ± 0.03 a
18	ethyl 2-oxopropanoate	1258	531.71	1.15215	0.83 ± 0.02 c	1.1 ± 0.02 a	1.01 ± 0.02 b	0.54 ± 0.03 d
19	2-pentyl furan	1242.9	505.377	1.2561	6.00 ± 0.08 c	6.95 ± 0.12 b	8.56 ± 0.09 a	7.07 ± 0.11 b
20	(E)-2-hexen-1-al-D	1234.3	490.965	1.18403	0.93 ± 0.01 c	1.38 ± 0.00 a	1.34 ± 0.02 b	0.68 ± 0.01 d
21	(E)-2-hexen-1-al-M	1233	488.964	1.52077	1.54 ± 0.01 b	1.70 ± 0.01 a	1.32 ± 0.00 d	1.42 ± 0.00 c
22	3-Methyl-1-butanol	1222.4	471.708	1.24493	0.59 ± 0.01 d	1.00 ± 0.01 c	1.31 ± 0.01 a	1.09 ± 0.01 b
23	Heptaldehyde-D	1199.4	436.609	1.33139	5.40 ± 0.04 a	3.76 ± 0.04 b	2.43 ± 0.02 d	3.34 ± 0.02 c
24	Heptaldehyde-M	1198.1	434.711	1.69727	3.31 ± 0.06 a	2.61 ± 0.03 c	1.99 ± 0.02 d	2.94 ± 0.01 b
25	1-Penten-3-ol	1174.9	400.919	0.94402	1.62 ± 0.02 b	1.68 ± 0.00 a	1.53 ± 0.01 c	1.65 ± 0.01 a
26	(E)-2-Pentenal-D	1148.6	365.148	1.10827	0.48 ± 0.00 c	0.74 ± 0.01 a	0.55 ± 0.01 b	0.44 ± 0.01 d
27	(E)-2-Pentenal-M	1148.2	364.655	1.36076	0.77 ± 0.01 c	0.91 ± 0.00 a	0.65 ± 0.00 d	0.8 ± 0.01 b
28	1-hexanal	1101.9	309.422	1.565	6.70 ± 0.34 b	7.49 ± 0.22 a	6.77 ± 0.36 b	7.77 ± 0.43 a
29	(E)-2-Butenal-D	1064	275.047	1.03763	0.11 ± 0.00 c	0.16 ± 0.01 b	0.18 ± 0.00 a	0.09 ± 0.00 d
30	(E)-2-Butenal-M	1063.7	274.826	1.20406	0.41 ± 0.02 a	0.46 ± 0.03 a	0.44 ± 0.05 a	0.35 ± 0.02 b
31	1-Propanol	1058.1	270.185	1.11579	0.20 ± 0.01 a	0.20 ± 0.01 a	0.19 ± 0.03 a	0.2 ± 0.02 a
32	2-methyl-1-propyl acetate	1029.3	247.536	1.23429	0.09 ± 0.01 a	0.07 ± 0.01 b	0.07 ± 0.00 b	0.08 ± 0.01 b
33	2-butanol	1018.6	239.669	1.15379	0.10 ± 0.01 ab	0.09 ± 0.01 b	0.06 ± 0.01 c	0.11 ± 0.01 a
34	2-Pentanone	997.1	224.498	1.36991	0.45 ± 0.01 c	0.46 ± 0.01 c	0.73 ± 0.02 a	0.58 ± 0.01 b
35	n-Pentanal	998.2	225.2	1.42766	4.99 ± 0.06 b	5.09 ± 0.08 b	4.43 ± 0.05 c	5.31 ± 0.11 a
36	3-methyl-2-pentanone	1009.3	232.926	1.47316	0.98 ± 0.03 c	1.02 ± 0.02 c	1.16 ± 0.01 a	1.12 ± 0.02 b
37	Ethanol-M	968.6	208.27	1.04506	3.57 ± 0.05 c	3.66 ± 0.05 b	3.28 ± 0.03 d	4.24 ± 0.04 a
38	Ethanol-D	951.2	199.043	1.12972	2.64 ± 0.02 b	2.75 ± 0.01 a	2.39 ± 0.01 c	2.77 ± 0.01 a
39	2-Butanone	916.4	181.727	1.25013	2.41 ± 0.09 a	2.02 ± 0.05 b	2.54 ± 0.06 a	2.48 ± 0.06 a
40	Acetic acid ethyl ester	930	188.287	1.3298	1.34 ± 0.02 b	1.26 ± 0.02 d	1.30 ± 0.01 c	1.44 ± 0.02 a
41	3-Methyl butanal	930	188.287	1.40745	0.47 ± 0.01 c	0.62 ± 0.02 b	0.81 ± 0.01 a	0.84 ± 0.02 a
42	Butanal	891.3	170.201	1.28591	0.64 ± 0.02 c	0.64 ± 0.02 c	0.98 ± 0.02 a	0.73 ± 0.02 b
43	2-propanone	834.2	146.618	1.11981	10.71 ± 0.15 b	10.08 ± 0.16 c	9.43 ± 0.16 d	11.08 ± 0.2 a
44	Propanal	808.3	137.043	1.14141	1.36 ± 0.05 b	1.35 ± 0.07 b	1.42 ± 0.04 b	1.65 ± 0.11 a
45	(Z)-furan linalool oxide-D	1414.2	793.365	1.26345	0.47 ± 0.01 c	0.51 ± 0.01 b	0.56 ± 0.01 a	0.27 ± 0.01 d
46	(Z)-furan linalool oxide-M	1412.4	790.188	1.81287	0.86 ± 0.02 c	1.26 ± 0.01 b	1.35 ± 0.00 a	0.80 ± 0.01 d
47	2-Octanone-D	1300.1	609.344	1.33623	0.19 ± 0.00 d	0.31 ± 0.01 b	0.86 ± 0.01 a	0.29 ± 0.01 c
48	2-Octanone-M	1300.8	610.349	1.75856	0.65 ± 0.01 d	0.76 ± 0.01 c	1.14 ± 0.02 a	0.84 ± 0.01 b
49	Ethyl hexanoate	1250.2	518.067	1.33512	0.25 ± 0.01 c	0.25 ± 0.01 c	0.30 ± 0.01 a	0.28 ± 0.01 b
50	3-Methyl-2-butenal-D	1215.7	461.283	1.09296	0.15 ± 0.00 a	0.13 ± 0.00 b	0.12 ± 0.00 c	0.14 ± 0.01 ab
51	3-Methyl-2-butenal-M	1215.1	460.4	1.36187	0.35 ± 0.01 b	0.33 ± 0.01 c	0.22 ± 0.01 d	0.38 ± 0.01 a
52	2-Heptanone	1193.3	427.742	1.63501	2.78 ± 0.01 c	3.80 ± 0.02 b	6.05 ± 0.15 a	3.93 ± 0.02 b
53	1-butanol-D	1159.9	380.051	1.18087	0.09 ± 0.01 b	0.10 ± 0.01 a	0.12 ± 0.01 a	0.08 ± 0.01 b
54	1-butanol-M	1158.9	378.762	1.37666	0.42 ± 0.01 b	0.43 ± 0.01 b	0.47 ± 0.01 a	0.47 ± 0.01 a
55	2-ethyl hexanol	1521.1	1015.946	1.41443	0.16 ± 0.02 c	0.23 ± 0.01 b	0.29 ± 0.01 a	0.12 ± 0.02 d
56	Decanal	1494.2	954.836	1.53898	0.35 ± 0.01 a	0.26 ± 0.01 b	0.23 ± 0.02 c	0.24 ± 0.02 c
57	(E.E)-2.4-heptadienal	1484.5	933.496	1.19206	0.11 ± 0.01 b	0.15 ± 0.01 a	0.10 ± 0.01 bc	0.08 ± 0.02 c
58	(E)-3-penten-2-one	1114.5	323.619	1.09344	0.16 ± 0.01 a	0.15 ± 0.00 a	0.12 ± 0.00 b	0.17 ± 0.00 a
59	2-methyl-1-Propanol	1112.3	321.042	1.17119	0.32 ± 0.00 b	0.27 ± 0.00 c	0.25 ± 0.00 d	0.38 ± 0.00 a

Notes: M and D suffixed after the chemicals indicated monomer and dimer, respectively. Different letters in the same row denote significant differences (*p* < 0.05).

## Data Availability

Data supporting the discovery of our work are available within the paper and its Appendix A.

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
