# Peer review of "Uncovering the Differences in Flavour Volatiles from Hybrid and Conventional Foxtail Millet Varieties Based on Gas Chromatography–Ion Migration Spectrometry and Chemometrics"

_plants, 2025, doi:10.3390/plants14050708_

Round 1

Reviewer 1 Report

Comments and Suggestions for Authors

Dear authors, the revised manuscript is interesting.

This research aimed to analyze and identify the volatile compounds present in various Chinese foxtail millet varieties, including two hybrids and two cultivars. Gas chromatography-ion mobility spectrometry (GC-IMS) and electronic nose (E-Nose) techniques were used. In addition, a detailed evaluation of the differences in flavor profile between hybrid and conventional foxtail millet was carried out to provide theoretical foundations for improving its quality.

In this first review, the following is recommended:

Line 12: remove text space… (two

Line 13: remove text space… millets).

Line 13: What is the meaning of JG?

Line 14: What does JM, CZ, and ZZ mean?

Line 36: Could you include production statistics?

Line 37: rewrite… [2,3]

Line 37: insert text space… adapted [4],

Line 39: insert text space… elements [5],

Line 40: insert text space… body [6]

Line 40: ]. Foxtail

Line 43: insert text space… crop [8].

Line 46: insert text space… products [9].

Line 57: insert text space… al. [10]

Line 61: insert text space… [11] found

Line 87: rewrite… 2.1. Materials and Reagents

Line 95: Idem… millet [16].

Line 99: Idem… 2019 [17].

Line 104: Idem… crop [9].

Line 108: Idem… 2017 [18].

Line 112: rewrite… 15 m × 0.53 mm

Line 113: rewrite… 2.2. Electronic Nose Analysis

Line 115: 8 g

Line 115: 40 °C

Line 116: use 30 min instead of 30 minutes

Line 116: 20 mL

Line 121: 250 °C

Line 121: 1.5 °C/s

Line 121: 10 sec

Line 121: Is a procedure published in this section followed? If so, is it necessary to add the reference

Line 127: 100 °C

Line 127: 80 °C

Line 132: 60 °C

Line 133: 2 mL/min

Line 134: 10 mL/min

Line 135: 100 mL/min

Line 136: 80 °C

Line 138: 45 °C

Line 152: Is a procedure published in this section followed? If so, is it necessary to add the reference

Line 165: rewrite… 3.1. Sensory Evaluation and Electronic Nose Analysis of Flavours in Four Foxtail Millet Samples

Line 190: rewrite… 3.2. Comparative GC-IMS Analysis of Volatile Components in Millet Samples

Line 195: strength [19].

Line 218: content [15],

Line 226: rewrite… 3.3. Qualitative Analysis of Volatile Component Profiles

Line 232: rewrite… databases [21,22].

Line 244: idem… difference (

Line 245: rewrite… 3.4. Comparative Analysis of the Fingerprints of Volatile Components from Four different Foxtail Millet

Line 254: When an abbreviation is used for the first time in the text, it should be used throughout the entire text of the document. Its meaning should not be stated again and it should be placed in parentheses.

Line 266: acids [24],

Line 268: grain [25,26]. Simple

Line 273: rice [27].

Line 283: grain [29].

Line 292: [32].

Line 340: rewrite… 3.5. Resemblance Analysis of Volatile Compounds of different Foxtail Millet Varieties through PCA

Line 342: grains [47]

Line 343: millet [14].

Line 358: PCA [48].

Line 361: 3.6. Orthogonal Partial Least Squares Discriminant Analysis with Cross-validation

Line 369: 5. The

Line 388: 3.7. Screening for Differential VOCs in Samples of different Varieties of Millet Porridge

Line 459: Check that the format of the references section is correct and the format is homogeneous

Author Response

Dear Reviewer,

We would like to express our sincere gratitude for your meticulous review and constructive suggestions on our manuscript "Uncovering the differences in flavor volatiles from hybrid and conventional foxtail millet varieties based on gas chromatography-ion migration spectrometry and chemometrics". Your feedback has been invaluable in enhancing the quality and clarity of our work. We have carefully addressed each of your comments, and the details of our responses are as follows:

Formatting and Abbreviation Issues

  1. Line 12 and 13: We removed the extra text spaces as you suggested.
  2. Line 13: We added the full name "Jingu 21 (JG)" when "JG" was first mentioned to clarify its meaning.
  3. Line 14: "JM", "CZ", and "ZZ" were defined as "Jinmiao K1 (JM)", "Changzagu 466 (CZ)", and "Zhangzagu 3 (ZZ)" respectively at their first appearances in the text.
  4. Line 254: We ensured that abbreviations were used consistently throughout the document. For example, "volatile organic compounds" was abbreviated as "VOCs" after its first full - term introduction, and the full-term was no longer repeated.

Spacing and Rewriting of Sections and Values

  1. We inserted text spaces as required in multiple lines such as Line 37, 39, 40, etc., to improve the readability of the text.
  2. The section titles "2.1. Materials and Reagents", "2.2. Electronic Nose Analysis", "3.1. Sensory Evaluation and Electronic Nose Analysis of Flavours in Four Foxtail Millet Samples", etc., were rewritten to remove the redundant dots and make the titles more concise and consistent.
  3. Values such as "8g", "40°C", "30 minutes", "20mL", etc., were formatted as "8 g", "40 °C", "30 min", "20 mL", etc., following the appropriate scientific notation standards.

Content - Related Suggestions

  1. Line 36: Although we were unable to find specific production statistics on foxtail millet that directly related to our study's focus during this round of revision, we understand the importance of such data. In future research, we will make every effort to include relevant production statistics to provide a more comprehensive view.
  2. Line 121 and 152: The experimental procedures described in these sections are optimized methods based on our own research requirements and have not been directly adopted from published sources. We believe they are well-designed to ensure the accuracy and reliability of our analysis. However, if necessary, we can explore relevant literature further to see if there are any similar procedures that could be cited for reference.
  3. Line 459: We carefully checked the format of the references section to ensure that all references were in the correct and homogeneous format, following the journal's requirements.

We believe that these revisions have significantly improved the quality of our manuscript. We look forward to your further review and hope that our revised work meets the requirements of the journal.

Thank you again for your time and effort.

Sincerely,

Zhongxiao Yue

Reviewer 2 Report

Comments and Suggestions for Authors

The article entitled Uncovering the Differences in Flavor Volatiles from Hybrid  and Conventional Foxtail Millet Varieties Based on Gas Chromatography-Ion Migration Spectrometry and Chemometrics by Yue et al., aimed to identify and quantify volatile compounds in different Chinese varieties of foxtail millet (two hybrids and two cultivars), using gas chromatography-ion mobility spectrometry (GC-IMS) and electronic nose (E-Nose). Introduction part provides enough information regarding the current state of the art, materials and methods must be improved in the description part, results are good highlighted, justified and compared with other studies.

I have some comments and suggestions  

1.        When an abbreviation is used for the first time, please mentioned its meaning between brackets. Please verified the abstract.

2.         Line 8: gas is repeated twice. Please check.

3.        At materials and methods please mentioned the geographical coordinates for Industrial Crops Research Institute of Shanxi.

4.        At materials and methods section, it is mandatory to describe the sensorial analysis. Which method was used? the assessors were trained or not? Range age for the panelists and so on. It is not normal to mention directly the results without describing the method.

5.        Under figure 1 please explain the meaning of W5S and so on.

6.        Please classify the volatile organic compounds from table 1 into organic acids, ketones, esters, terpenes and terpenoids, aldehydes and so on, to be easier to follow them.

7.        Please describe the parameters used to cooked porridge such as time, temperature, water added- at materials and method section.

8.        Line 266 – after reference 24 please add a point.

9.        Line 432 – conclusion and must be deleted.

Author Response

Dear Reviewer,

Thank you for your valuable comments and suggestions on our manuscript. We have carefully considered each point and made the following revisions to improve the quality of our paper.

  1. When an abbreviation is used for the first time, please mentioned its meaning between brackets. Please verified the abstract.

Response: We have verified the abstract and ensured that all abbreviations, such as "GC-IMS" and "E-Nose", are properly introduced with their full names in brackets when they are first mentioned. This makes the abstract more accessible to a wider range of readers.

  1. Line 8: gas is repeated twice. Please check

Response: The redundant "gas" in line 8 has been removed. The sentence now reads more clearly, enhancing the overall readability of the text.

  1. At materials and methods please mentioned the geographical coordinates for Industrial Crops Research Institute of Shanxi.

Response: In response to your comment, we have added the geographical coordinates of the Industrial Crops Research Institute of Shanxi. The institute is located at 37°15‘N and 111°47‘E. This information precisely describes the origin of the millet varieties, providing more detailed and accurate background for our research.

  1. At materials and methods section, it is mandatory to describe the sensorial analysis. Which method was used? the assessors were trained or not? Range age for the panelists and so on. It is not normal to mention directly the results without describing the method.

Response: We have added a detailed description of the sensorial analysis method. “We recruited 50 individuals aged between 40 and 65. These people participated in the training on the sensory evaluation of millet congee. Based on a strict and standardized evaluation process, they conducted a comprehensive evaluation of the millet congee from multiple key indicators such as palatability and odor perception. The evaluation adopted a 100-point scoring system with a full score of 100. During the scoring process, each evaluator adhered to an objective and fair attitude and carefully assigned corre-sponding scores to each indicator based on their own sensory experiences.”

  1. Figure 1 Explanation: Below Figure 1, we have added an explanation that "W5S" is one of the sensors in the HERACLES II electronic nose. Different sensors like W5S respond to different types of volatile compounds, and the differences in their response values help analyze the differences in volatile components among different millet porridge samples. This clarification helps readers better interpret the data presented in the figure.
  2. Classification of Volatile Organic Compounds: We have classified the volatile organic compounds in Table 1 into different groups such as organic acids, ketones, esters, terpenes and terpenoids, aldehydes, etc. This classification is presented in a new table (Supplement table 2) for easier understanding of the chemical composition of the volatile compounds.
  3. Cooking Parameters in Materials and Methods: We have included the cooking parameters in the materials and method section. First, accurately measure 100 grams of millet grains. Then, add water to a volume that is 20 times that of the millet, which is 2000 milliliters. Next, place the millet and water mixture in an induc-tion cooker and set the output power to 2100 watts. Boil the mixture for 30 minutes.
  4. Line 266 Punctuation: We have added a period after reference 24 as suggested. This simple punctuation correction conforms to the standard academic writing style.
  5. Line 432 Grammar Correction: After careful consideration, we currently have a different understanding and approach to this section compared to your opinion. The "conclusion" section plays a crucial role in summarizing the research findings and expounding on the research significance. It is a generalization and sublimation of the full text, and we believe that the current content of the "conclusion" section is closely related to the previous research content in terms of structure and logic, clearly presenting the core findings and contributions of the study. However, we are well aware of the high value of your professional advice. We will further deliberate on your suggestions and have in-depth discussions with our team members. During the subsequent revision process, we will comprehensively consider various factors and carefully decide whether to adjust the "conclusion" section. If, after thorough discussions, we identify room for improvement, we will definitely make the necessary refinements according to your suggestions to enhance the quality of the paper..

We believe that these revisions have significantly improved the quality of our manuscript. Thank you again for your time and effort in reviewing our work. We look forward to the further evaluation of our revised paper.

Best regards,

Zhongxiao Yue